# Genetic History of the Altai Breed Horses: From Ancient Times to Modernity

**DOI:** 10.3390/genes14081523

**Published:** 2023-07-26

**Authors:** Mariya A. Kusliy, Anna A. Yurlova, Alexandra I. Neumestova, Nadezhda V. Vorobieva, Natalya V. Gutorova, Anna S. Molodtseva, Vladimir A. Trifonov, Kseniya O. Popova, Natalia V. Polosmak, Vyacheslav I. Molodin, Sergei K. Vasiliev, Vladimir P. Semibratov, Tumur-O. Iderkhangai, Alexey A. Kovalev, Diimaajav Erdenebaatar, Alexander S. Graphodatsky, Alexey A. Tishkin

**Affiliations:** 1Department of the Diversity and Evolution of Genomes, Institute of Molecular and Cellular Biology SB RAS, 630090 Novosibirsk, Russia; alexn@mcb.nsc.ru (A.I.N.); vorn@mcb.nsc.ru (N.V.V.); rada@mcb.nsc.ru (A.S.M.); vlad@mcb.nsc.ru (V.A.T.); popova.ksenia@mcb.nsc.ru (K.O.P.); graf@mcb.nsc.ru (A.S.G.); 2Laboratory of Genomics, Department of Regulation of Genetic Processes, Institute of Molecular and Cellular Biology SB RAS, 630090 Novosibirsk, Russia; 3Department of Human Molecular Genetics, Institute of Cytology and Genetics SB RAS, 630090 Novosibirsk, Russia; kras_nv@bionet.nsc.ru; 4Paleometal Archeology Department, Institute of Archaeology and Ethnography SB RAS, 630090 Novosibirsk, Russia; polosmaknatalia@gmail.com (N.V.P.); molodin@archaeology.nsc.ru (V.I.M.); svasiliev@archaeology.nsc.ru (S.K.V.); 5Department of Archaeology, Ethnography and Museology, Altai State University, 656049 Barnaul, Russia; svp9039116217@mail.ru (V.P.S.); tishkin210@mail.ru (A.A.T.); 6Department of Archaeology, Ulaanbaatar School, National University of Mongolia, 13343 Ulaanbaatar, Mongolia; iderkhangai83@gmail.com (T.-O.I.), ediimaajav@gmail.com (D.E.); 7Department of Archaeological Heritage Preservation, Institute of Archaeology of the Russian Academy of Sciences, 117292 Moscow, Russia; chemurchek@mail.ru

**Keywords:** ancient DNA, mitochondrial DNA, Altai horse, phylogenetic analysis, population genetic analysis

## Abstract

This study focuses on expanding knowledge about the genetic diversity of the Altai horse native to Siberia. While studying modern horses from two Altai regions, where horses were subjected to less crossbreeding, we tested the hypothesis, formulated on the basis of morphological data, that the Altai horse is represented by two populations (Eastern and Southern) and that the Mongolian horse has a greater genetic proximity to Eastern Altai horses. Bone samples of ancient horses from different cultures of Altai were investigated to clarify the genetic history of this horse breed. As a genetic marker, we chose hypervariable region I of mitochondrial DNA. The results of the performed phylogenetic and population genetic analyses of our and previously published data confirmed the hypothesis stated above. As we found out, almost all the haplotypes of the ancient domesticated horses of Altai are widespread among modern Altai horses. The differences between the mitochondrial gene pools of the ancient horses of Altai and Mongolia are more significant than between those of modern horses of the respective regions, which is most likely due to an increase in migration processes between these regions after the Early Iron Age.

## 1. Introduction

The migration routes of many ancient peoples passed through the territory of Altai, and the most populous of them were associated with the use of domestic horses for transportation. Since the early stages of their domestication, Altai horses have been of great importance in various areas of society: economic, political, spiritual, and social [1]. The crucial role of the horse in the spiritual culture of the Altai people can clearly be seen in their folk beliefs and rituals, folklore, and fine arts [2]. The Altai breed belongs to the native breeds of Siberia, which were subjected to interbreeding to a lesser extent than farm breeds. At the same time, modern horses from different regions of Altai differ from each other in appearance; Southern Altai horses are larger than horses from the Eastern Altai. Researchers correlate this with their origin and differences in feeding conditions, suggesting that the Eastern Altai horses are genetically closer to the Mongolian horse [3]. Despite the high significance of the horse in the life of the Altai people, there have been a relatively small number of molecular genetic studies carried out that could shed light on the origin of the Altai breed and its relationship with others. Mitogenome and genome haplotypes were identified for some horses of the Biyke culture (end of the 9th to 2nd–3rd quarter of the 6th centuries BCE) in the Altai Mountains of Russia [4], and for horses of the Pazyryk culture (second half of the 6th to 2nd centuries BCE) in the Altai Mountains of Eastern Kazakhstan and Northwestern Mongolia [5,6,7,8,9]. These data indicated that horses of the Biyke culture of Altai have a lower degree of genetic relationship with horses of the Khirigsuurs and “Deer” Stone culture of Mongolia in the transition period from the Bronze Age to the Early Iron Age [4] than horses of the Pazyryk culture of the Mongolian Altai and the Xiongnu culture of Mongolia (from the middle and end of the same period, respectively) [8,9]. This may be due to the peculiarities of contact between the cultures under consideration and their migration routes. The study of a control region of mitochondrial DNA in 11 modern Altai horses from Russia showed higher genetic diversity of this sequence in horses of different breeds from the Baltic region than in horses of the Altai breed [10], while studies of microsatellite sequences in Altai horses and horses of other native breeds of Siberia and Mongolia revealed the largest diversity in the former with the presence of unique alleles [11,12,13]. Since comparison of the mitochondrial genetic diversity of the Altai horse with other native breeds of the region has not yet been implemented, and the domestic horses of the Altai Mountains of Russia are underrepresented in genetic studies compared to horses of the Altai Mountains of neighboring countries, we decided to close this gap. We investigated the genetic composition of modern domestic horse groups from two farms in the Altai region of Russia, tested the hypothesis of the genetic proximity of modern horses of the Eastern Altai and Mongolia [3], and identified common haplotypes among modern, medieval and ancient Altai horses from various archaeological sites in the region under study. For this purpose, we investigated the hypervariable region I of the mitogenome. The analysis of this genetic marker was used to determine the levels of inbreeding and selection pressure [14], the degree of genetic diversity in horse breeds [15], and differences in haplotype diversity between different populations of the same horse breed [16,17], as well as between gene pools of various horse breeds [14,17,18,19]. A similar analysis was carried out for horses of Kazakh and Mongolian breeds from territories adjacent to Altai, which showed similarities between the mitochondrial gene pool of ancient and modern Kazakh horses of Kazakhstan and China [6,20] and that of modern and ancient horses of Mongolia [5,13,21]. The above studies reveal the effectiveness of using the selected genetic marker for the tasks we had identified. Our study determined maternal genetic proximity, a similar degree of haplotype diversity in modern horses of the Eastern Altai and Mongolia, and a greater difference between the mitochondrial gene pools of ancient Altai horses and those of modern ones, which probably indicates an increase in the interconnection of archaeological cultures and peoples of the considered regions after the Early Iron Age.

## 2. Materials and Methods

### 2.1. Information about the Samples

We studied hair follicles and hair fragments of 29 and 81 modern horses of local groups from private and gene pool (Chingiz) farms in Ulagansky District (Eastern Altai), respectively, and 18 modern horses of the local group of a private farm in Kosh-Agachsky District (Southeastern Altai), all located in the Altai Republic (Russia). In the text below, we will refer to the groups of horses from Ulagansky District as the Ulagan horses, and the group of horses from Kosh-Agachsky District as the Kosh-Agach horses. It should be noted that the gene pool group consists only of horses of the Altai breed, and “mixed” groups also include hybrids with horses of other breeds. The studied ancient specimens were bones and teeth and belonged to eight ancient horses from six archaeological sites in Altai and nine ancient horses from three archaeological sites in Mongolia. The sequences of the studied mitogenome control region fragment for samples DC, AkX-X, Ver1-1, Gan-X, and Er-1 were taken from articles previously published by the co-authors of this article [22,23]. Information on the geographical origin and dating of the used bone samples of the ancient horses is given in Appendix A, as well as in Figure 1, which also shows the locations of the above Altai horse farms.

### 2.2. Modern Sample Mitogenome Hypervariable Region I Sequencing

All stages of the modern sample experiments of the present study were carried out in a modern DNA facility at the Department of the Diversity and Evolution of Genomes of the Institute of Molecular and Cellular Biology of the Siberian Branch of the Russian Academy of Sciences (IMCB SB RAS) (Novosibirsk, Russia). All experiments were approved by the Ethics Committee on Animal and Human Research at the IMCB SB RAS, Russia (Protocol №1 from 29 December 2022). 

DNA isolation from modern samples was performed based on the user-developed method of purification of total DNA from nails, hair, or feathers (Qiagen 2014) using the DNeasy Blood and Tissue Kit (Qiagen, Hilden, Germany). A 594 bp mitogenome control region fragment was obtained via PCR using AmpliTaq Gold^®^ DNA Polymerase, PCR Gold Buffer, primers B10 (5′-ACC ATC AAC ACC CAA AGC T-3′) and F10 (5′-CTT TGA CGG CCA TAG CTG AGT-3′), extracting modern sample DNA (60–90 ng per reaction) with standard AmpliTaq Gold^®^ PCR master mix content and reaction conditions (AmpliTaq Gold^®^ DNA Polymerase Protocol, Thermo Fisher Scientific, Vilnius, Lithuania). Purification of the obtained PCR products was carried out using ExoSAP-IT™ PCR Product Cleanup Reagent (Thermo Fisher Scientific) according to the manufacturer’s protocol. Sequencing reactions were performed using the BigDye™ Terminator v3.1 Cycle Sequencing Kit (Thermo Fisher Scientific) based on the manufacturer’s protocol. Sequencing reaction products were purified using the BigDye XTerminator ™ Purification Kit (Thermo Fisher Scientific) according to the manufacturer’s instructions. Capillary electrophoresis was carried out using a Genetic Analyzer Sanger Sequencing 3500 Series, Applied Biosystems (Thermo Fisher Scientific), at the Molecular and Cellular Biology core facility of the IMCB SB RAS, Novosibirsk, Russia.

### 2.3. Ancient Sample Mitogenome Hypervariable Region I Sequencing

All stages of the ancient sample experiments of the present study up to the PCR stage were carried out in the ancient DNA facility at the Department of the Diversity and Evolution of Genomes of the IMCB SB RAS (Russia) in accordance with the main “criteria of authenticity” for aDNA research [24]. DNA from ancient samples was obtained using the protocol of Yang and co-authors [25] with minor changes: lysis time was reduced by using (NH_4_)_4_EDTA [26]; sodium dodecyl sulfate was replaced with sodium lauroyl sarcosinate in the lysis buffer; the sample incubation stage in the lysis buffer at 37 °C was absent; MinElute spin columns (Qiagen) were used instead of QIAquick spin columns (Qiagen) during the purification step to extract shorter DNA fragments. In addition, the preparation of bone powder included pre-treatment of the sample surface from all sides with ultraviolet irradiation for 30 min. 

Overlapping fragments of the mitogenome control region (150 bp and 177 bp in size) were obtained via PCR using Phusion™ High–Fidelity DNA Polymerase and Buffer, primers Con1_F_ (5′-TCT TCC CCT AAA CGA CAA CAA-3′), Con1_R_ (5′-GCT TAT TAT TCA TGG GGC AGA C-3′), Con2_F_ (5′-GAA TGG CCT ATG TAC GTC GTG-3′), Con2_R_ (5′-TGG TGA TTA AGC TCG TGG AAC-3′), extracting ancient sample DNA (30–50 ng per reaction) with standard Phusion High-Fidelity PCR master mix content and reaction conditions (Phusion High-Fidelity DNA Polymerase User Guide, Thermo Fisher Scientific) with one modification, which was to add bovine serum albumin (400 ng/μL concentration in PCR mix) to the reaction mixture. The selection of primers for the region under study for ancient DNA experiments was carried out in accordance with the main characteristic of ancient DNA, high fragmentation [27]. Each PCR was repeated one time. Purification of the obtained PCR products was performed using the MinElute PCR Purification Kit (Qiagen) based on the manufacturer’s protocol with the following changes: double purification of the sample with Buffer PE (wash buffer, Qiagen); incubation of the sample in Buffer EB (elution buffer, Qiagen) with 0.05% Tween 20, preheated to 37 °C, for 10 min at 37 °C; performing the first centrifugation at 8000 g, and the last one for several minutes. 

Based on the purified PCR products, single- or double-indexed paired-end libraries were prepared using the TruSeq^®^ Nano DNA Sample Preparation Kit reagents (Illumina, San Diego, CA, USA) and the manufacturer’s instructions with the following modifications: DNA fragmentation was not performed; 250 ng of PCR products was used; the first purification of the library fragments was carried out using the MinElute PCR Purification Kit (Qiagen), and the second and third purifications included the addition of one volume of Sample Purification Beads (Illumina) to the sample; all elution steps were performed in EB buffer with 0.05% Tween 20; library amplification was carried out for 12 cycles. The amplified libraries were quantified using primers A1 (5′-AAT GAT ACG GCG ACC ACC GAG ATC T-3′) and A2 (5′-CAA GCA GAA GAC GGC ATA CGA GAT-3′) complementary to the Illumina adapter conserved fragments in 2.5X Reaction mixture containing SYBR Green I and ROX fluorescent dyes (Synthol) and according to the manufacturer’s protocol. The normalized and equimolar pooled libraries constituted a library pool that was sequenced on a MiSeq Sequencer (Illumina, San Diego, CA, USA) using the MiSeq Reagent Kit v2 (Illumina, San Diego, CA, USA) (300 cycles, 2 × 150 bp) according to the manufacturer’s protocol (MiSeq System Guide, Illumina) at the Molecular and Cellular Biology core facility of the IMCB SB RAS, Russia.

### 2.4. Secondary Sequencing Data Analysis

The sequences of the 246 bp mitogenome control region fragment for modern horses were obtained using Sequencing Analysis Software v7.0 (primary analysis tool, Thermo Fisher Scientific) and Sequence Scanner Software v2.0 (viewer, Thermo Fisher Scientific).

For ancient and medieval horses, the sequences of the 246 bp mitogenome control region fragment were obtained using PALEOMIX BAM Pipeline v1.3.2 [28]. Removal of adapter sequences and collapse of reads was performed using AdapterRemoval v2.2.2 [29]. Sequence aligner bwa v0.7.15 [30] was used to perform collapsed read alignment against the reference sequence of the horse mitogenome (GenBank accession №: NC_001640.1), with a minimum read mapping quality of 25. The MapDamage v2.2.0 computational framework [31] helped recalculate base quality scores according to the probability of post-mortem DNA damage at each position in the sequence. The bioinformatics software platform Geneious Prime v2020.2.4 (https://www.geneious.com; Biomatters Ltd., Auckland, New Zealand) was used to visualize the resulting alignment and obtain a consensus sequence of the studied fragment of the mitogenome control region in which the quality of each base would be higher than 60% of the total adjusted base quality.

### 2.5. Genetic Diversity Analysis

We determined haplotypes of the mitogenome control region according to the classification of Cieslak and co-authors [7]. 

Genetic diversity was identified within and between the studied groups of horses. The calculation of F_ST_ values (evolutionary Nei’s genetic distance, 10,000 permutations), which characterize the degree of group differentiation, genetic and haplotype diversity, as well as molecular inter-population and intra-population variance (AMOVA (Analysis of MOlecular VAriance), 1023 permutations) within predetermined groups, was performed in the integrated software package Arlequin v3.5.2.2 [32]. Graphic visualization of the obtained values was performed using the R-function (pairFstMatrix.r) of the Arlequin v3.5.2.2 software package [32]. 

A phylogenetic median-joining network was constructed using Network v5.0.1.1 software (Fluxus Technology Ltd., Colchester, England; http://www.fluxusengineering.com), based on alignment of the 246 bp mitogenome hypervariable region I fragment sequences of modern, medieval and ancient horses of Altai and adjacent territories obtained here and in the study of Cieslak and co-authors [7].

The calculated haplogroup frequencies (shown in Appendix A) were used to perform a principal component analysis using jalview v2.11.0 software [33].

### 2.6. Data Availability

All sequences of the mitogenome control region fragment of the ancient, medieval, and modern horses of Altai and Mongolia that we studied here were submitted to the GenBank nucleotide sequence database under the registration numbers given in Appendix A (ancient samples) and Appendix A (modern samples).

## 3. Results

### 3.1. Haplotype Frequencies in the Studied Horse Groups

Mitogenome hypervariable region I haplotyping of modern horses of Altai from three groups (the Kosh-Agach “mixed” group and the Ulagan “mixed” and gene pool groups) revealed 19 haplotypes (10 haplogroups). The same analysis for ancient and medieval horses of Altai and Mongolia identified eight and six haplotypes (six and five haplogroups), respectively. Moreover, we discovered four new, previously unrecognized haplotypes in modern Altai horses, which is consistent with data from other researchers on the unique microsatellite alleles of the Altai horse breed [11,12,13]. The identified haplotypes of modern Altai horses and ancient and medieval horses of Altai and Mongolia are shown in Table 1, Appendix A, respectively.

Differences in haplotype composition between the studied groups of modern, ancient and medieval horses of Altai and Mongolia, as well as previously studied groups of ancient and modern horses of Altai and Mongolia, were visualized in the form of a phylogenetic network (Figure 2). From the already published sequences, we used 225 modern and 3 ancient mitogenome sequences of Mongolian horses ([7,13,17,34,35], KF197850.1–KF197940.1 GenBank accession numbers), and 52 modern and 11 ancient mitogenome sequences of Altai horses ([7,10], KF197142.1–KF197182.1 GenBank accession numbers).

Differences in haplogroup frequencies between the considered groups of horses are shown in Figure 3.

The data in Table 1 and Figure 3 indicate that haplogroups K, K3, and X2 are the most common among modern Altai horses, while, as can be seen from Figure 3, haplogroups K and X2 have the highest frequency in the Mongolian horse breed as well. Comparing the haplotype structures of all studied groups of modern Altai horses (Table 1, Figure 2), we can conclude that K2 and K3 haplotypes are the most widespread in both groups from Ulagan, and the haplotypes that have the highest frequency of occurrence in the Kosh-Agach group (B1, X2, X2 + 521) are less common in the Ulagan groups. When analyzing only the composition of haplotypes and not their frequency, greater degrees of genetic proximity between the Ulagan “mixed” and gene pool groups (7 common haplotypes) and between the Ulagan gene pool group and the Kosh-Agach “mixed” group (7 common haplotypes) are revealed compared to the Ulagan and Kosh-Agach “mixed” groups (5 common haplotypes). Figure 2 and Table 1 also show that haplotypes D and I are present only in the gene pool of the Ulagan “mixed” group, while only the X2 + 521 haplotype distinguishes the Kosh-Agach group from the Ulagan groups. 

As can be seen from the phylogenetic tree (Figure 2) and the diagram of haplogroup frequencies (Figure 3), except for the haplotype of the Paleolithic horse from Denisova Cave from the haplogroup of an extinct lineage of horses (haplogroup X11), all other haplotypes of the studied ancient (Early Iron Age) and medieval horses of Altai, or haplotypes closest to them, are often found among modern horses of Altai. Many identified haplotypes of ancient horses of Mongolia (Late Bronze–Early Iron Ages) are common both among Mongolian horses and among horses of Altai.

Phylogenetic reconstructions showed the presence of common haplotypes between horses from the Pazyryk (Ak1-1) and Turkic (Bi4) cultures of Altai, as well as between horse groups from Khirigsuurs and “Deer” Stone culture sites of Mongolia (Ush-8, Gan-18) and Pazyryk culture sites of Northern Altai (Ber2).

### 3.2. Statistical Population Genetics Analysis

The level of genetic (nucleotide, haplotype) diversity was calculated for all studied groups of horses. These data are shown in Table 2.

Based on the data in Table 2, among the combined ancient groups of horses, the group of Altai horses has the highest level of mitochondrial genetic diversity. Comparing the united published and studied groups of modern horses of Altai and Mongolia, we can conclude that the Ulagan “mixed” horse group (MAUM) and the Kosh-Agach horse group have the lowest degree of genetic diversity. Among the other groups, the united group of modern Mongolian horses (published data) has the highest genetic diversity, while the “mixed” group of Ulagan horses and the united group of Altai horses (published data) have the same level of haplotype diversity.

After determining the level of genetic diversity of the horse groups studied, we identified the degree of genetic similarity between them. The analyses described below were performed only on data from modern horses, since the statistical methods used can give unreliable values when there are large temporal differences in the data [36].

To achieve the goal described above, we carried out the following analyses: principal component analysis, determination of the F_ST_ values (Wright’s fixation index), and analysis of molecular variance (AMOVA).

The results of the principal component analysis performed on the basis of the identified haplogroup frequencies in the studied groups of horses (Appendix A) are presented in the form of a biplot (Figure 4). This analysis was performed to obtain preliminary results on differences in genetic diversity between the studied groups of horses. The results obtained were confirmed by other statistical analyses described below with a higher degree of reliability.

The presented two-dimensional plot (Figure 4) is based on the first two factors (F1, F2). It is statistically significant because the squared cosines of the variables associated with the above factors are high (Appendix A), and this plot shows ~70% of the variability in the data. 

Figure 4 indicates that using the first and second principal components, the groups closest to each other are the Kosh-Agach group (MAKM) and the Ulagan gene pool group (MAUG), as well as the united group of horses of the Altai breed (MAP) and the Ulagan “mixed” horse group (MAUM). Using the first principal component, which characterizes a much larger percentage of genetic diversity than the second, the group of Mongolian horses (MMP) is closer to the united group of Altai horses and the “mixed” group of Ulagan horses than to the other groups. At the same time, according to the constructed graph, it can be seen that the MAKM and MMP groups are maximally distant from each other.

The calculated F_ST_ values for all groups of modern Altai horses studied here show that the closest groups are MAUG and MAUM (F_ST_ = 0.02012), MAUG and MAKM (F_ST_ = 0.07008) are more differentiated, and the highest level of differentiation is observed between MAKM and MAUM (F_ST_ = 0.08262). At the same time, the *p*-value for the F_ST_ of the first two groups (MAUG and MAUM) confirms that the genetic differentiation between them is insignificant. As for the level of differentiation compared to the Mongolian horse group, it is moderate (F_ST_ = 0.07536) with the Kosh-Agach horse group, and insignificant with the Ulagan horse groups.

To determine the presence of a genetic structure through AMOVA, we compared groups of modern Altai horses united on the basis of common geographic location (Ulagansky or Kosh-Agachsky District) or breeding strategies (gene pool or “mixed” groups). It has been shown that the greatest percentage of genetic variance is accumulated within groups (Appendix A), and these differences are statistically significant. Statistically significant differences were also identified between the “mixed” Ulagan and Kosh-Agach groups (Appendix A: genetic variance is 9.61%), and between the Ulagan gene pool group and the Kosh-Agach “mixed” group (Appendix A: genetic variance is 7.21%). The smallest genetic differences were found between the Ulagan groups (Appendix A: genetic variance is 1.8%), with *p*-value range showing that these differences are insignificant. The presence of a genetic structure was determined only for the association of the Kosh-Agach and Ulagan united groups (Appendix A: genetic variance is 5.71%), while the genetic differences between these united groups are not significant.

## 4. Discussion

The obtained values of haplotype and nucleotide diversity level (Table 2) showed that the mitochondrial gene pool of the ancient horses of Altai is more diverse compared to the gene pool of the ancient horses of Mongolia. The revealed degree of mitochondrial genetic diversity in the united modern Altai horse group and the Ulagan “mixed” horse group are comparable and higher than that for the Ulagan gene pool group and the Kosh-Agach group, which are less diverse in mitotypes. This is most likely due to these horse groups having been less crossed with horses of other breeds. The greater haplotype diversity of the modern Mongolian horse group is highly likely to be associated with the same process. Based on the literature data [10], it can be concluded that, in terms of the level of diversity of the mitochondrial gene pool, modern horses of the Altai breed are comparable with Arabian horses (~0.88) and have a lower mitochondrial genetic diversity than modern Mongolian horses, whereas the ancient horses of Altai were more genetically diverse than the ancient horses of Mongolia. These changes over time are due to the peculiarities of the genetic history of both breeds (contact between associated cultures and breeding strategies). 

Based on the mitotype frequencies in the studied groups of modern Altai horses (Table 1), we revealed a greater degree of genetic diversity in the Ulagan groups of horses (“mixed” and gene pool) compared to the “mixed” group of Kosh-Agach horses. However, this conclusion is preliminary due to the small sample size of the Kosh-Agach horses; more horses from this region need to be examined to confirm this statement. The identified haplotype composition of the horse groups also indicates a greater proximity between the Ulagan “mixed” and gene pool groups than between each of them and the Kosh-Agach “mixed” group. The reason for this may be the isolation of these groups of horses in different regions of the Altai Mountains. At the same time, the main shared genetic diversity between the Kosh-Agach and Ulagan groups is associated with the Altai gene pool horse group in which the original gene pool of the studied breed is concentrated. In addition, some mitogroups found in the Ulagan “mixed” group were not identified in other studied groups of modern Altai horses, which may be due to crossbreeding of horses in this group.

Determination of the mitogenome hypervariable region I haplotypes in the studied Altai ancient and medieval horses showed that the Paleolithic horse from Denisova Cave (Northwestern Altai, DC horse) belongs to the haplogroup (X11), which, according to our and literature data [7,10], is not found among modern horses and before our study was discovered only in pre-Holocene horses from Northeastern Siberia. Horses of this mitogroup do not seem to have contributed to the formation of domestic horse herds. 

If we consider the studied Holocene horses from different archaeological cultures, we can see that the Biyke culture horses of the Southern Altai (Ak2-4, Ak2-5, Ak2-6) belong to the mitotypes (D2, K, K2) characteristic of Altai and adjacent territories in the Bronze and Iron Ages [7]. Turning to the Pazyryk culture, it can be noted that the horses of the Southern Altai in Russia belonging to this culture (Ak1-1, Ver1-1) have mitotypes (I, K + 601) that are widespread in the territories adjacent to Altai [7]. Horses of the Pazyryk culture from the Southern Altai on the territory of Kazakhstan were assigned to mitotypes A, B1, D2, D3, I, K3b, X2, and X4a, characteristic of this and adjacent regions at that time [7]. Among the horses of this culture from the Northern Altai (Ber2, Bi3), there are both haplotypes (X2b) common in synchronous and earlier (the Tuvan Aldy-Bel culture) sites of the region under consideration, and more typical ones for Eastern Europe and Asia Minor in antiquity (G1) [7]. The data obtained confirm the results of previous studies on the genetic diversity of the ancient horses of Altai and nearby territories [5,6,7] and also indicate possible migration paths from Asia Minor to Altai during the Pazyryk period [37]. Another, more characteristic, mitotype for Eastern Europe and Asia Minor (I) [7] was determined by us in a medieval horse of a Turkic culture (Bi4) from the Northern Altai, which corresponds to the migration of Turkic culture from the west [38]. Most haplotypes of the studied ancient Mongolian horses are typical for this region and time [7], while the mitochondrial gene pools of ancient Mongolian and Altai horses have a small number of intersections: they have only three common haplogroups (B, D, X2) and only four shared haplotypes (B1, D2, X2, X2b). These differences are presumed to be associated with the small number of contacts between the cultures of these regions in the Early Iron Age. The assumption of different origins should be confirmed by genome-wide studies.

Visualization of the haplotype compositions of the ancient and modern horse groups from the studied region (Figure 2) highlighted a small number of intersections between the mitochondrial genetic diversity of ancient Mongolian and Altai horses, and identified the common mitotype between the Pazyryk and Turkic horses of Altai. These phylogenetic data also showed the widespread prevalence of the ancient horse haplotypes of Altai and Mongolia among modern horses from these regions, and a greater intersection between the mitochondrial gene pools of modern Altai and Mongolian horses compared to the ancient ones. The revealed features indicate a possible succession of horses from the Pazyryk and Turkic cultures of Altai, and an increase in contact between the cultures of Altai and Mongolia after the Early Iron Age.

Contrasting six mitotypes of Ulagan horses, characteristic of Mongolian horses in the Bronze–Iron Ages, with three similar mitotypes of Kosh-Agach horses, as well as eight mitotypes of Ulagan horses from haplogroups common among modern Mongolian horses with six similar mitotypes of Kosh-Agach horses, testifies in favor of greater genetic similarity of the Mongolian horses with the horses of Ulagan (Eastern Altai) than with the horses of Kosh-Agach (Southern Altai). 

The population genetic analyses carried out (principal component analysis, F_ST_, AMOVA) confirmed the presence of two populations of horses of the Altai breed that live in geographically isolated areas: a mountain population formed in conditions of semi-free living in the subalpine and alpine high-mountain meadows of Ulagan and a steppe population formed in the semi-desert steppe conditions of Kosh-Agach. Based on analysis of the mitochondrial DNA control region, we showed that representatives of these populations have not only phenotypic differences [3] but also genetic differences. Differences in these populations may be due to the different needs of the inhabitants of these regions: in the mountains, the advantage is given to a drier and stronger horse that could easily move with riders or cargo over rocky terrain; in the steppe region, horses are more often used to obtain koumiss and meat, and therefore their selection was carried out in different directions.

The greater genetic similarity of Ulagan horses to Mongolian horses could be due to the greater degree of cross-breeding of Kosh-Agach horses with horses of farm breeds or with the peculiarities of their origin, since the ancient horse populations of Altai and Mongolia have significant differences in the gene pool.

Thus, our study opened up the prospects for further study of Altai horse populations from different regions, including by using other genetic markers.

## Figures and Tables

**Figure 1 genes-14-01523-f001:**
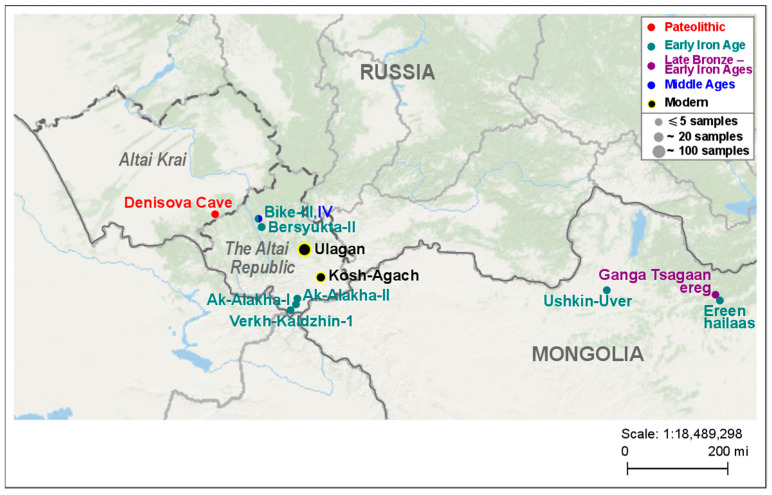
Geographic location and temporal affiliation of archaeological sites and farms in Altai and adjacent territories, horses from which were studied here. Different colors represent various time periods. The sizes of circles reflect the number of samples examined.

**Figure 2 genes-14-01523-f002:**
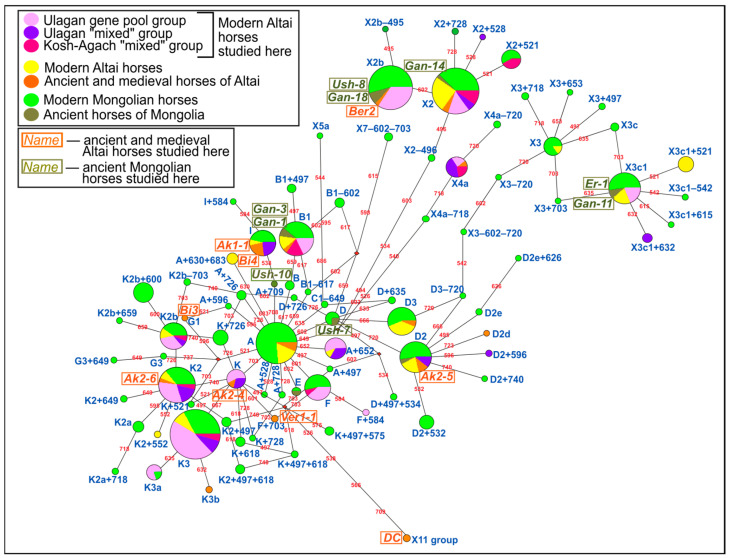
Phylogenetic median-joining network based on mitogenome hypervariable region I fragment sequences from modern, medieval and ancient horses of Altai and Mongolia. The size of the circle reflects the number of horses with this haplotype. The circle sector color indicates the geographic origin and temporal belonging: light green shows modern Mongolian horses, dark green indicates ancient Mongolian horses, yellow represents modern Altai horses, orange is associated with ancient and medieval Altai horses. The modern Altai horses investigated here are shown in dark pink (Kosh-Agach “mixed” group), violet (Ulagan “mixed” group), pink (Ulagan gene pool group). The ancient and medieval horses of Altai and ancient horses of Mongolia studied by us are highlighted in orange and green boxed haplotype names, respectively. Numbers on the network branches show the location of genetic variants from the beginning of 15,000 bp in the horse mitogenome reference sequence (NC_001640.1).

**Figure 3 genes-14-01523-f003:**
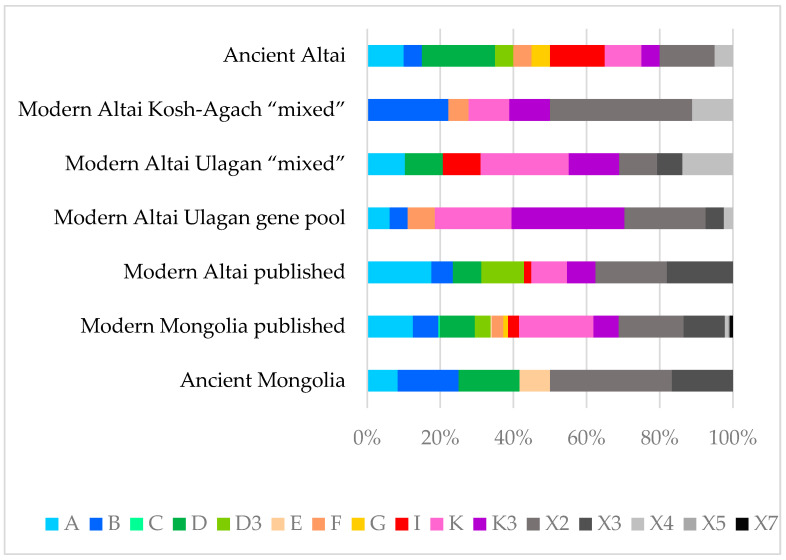
Diagram of the mitogenome hypervariable region I haplogroup frequency ratio between the studied horse groups. Different colors represent various haplogroups.

**Figure 4 genes-14-01523-f004:**
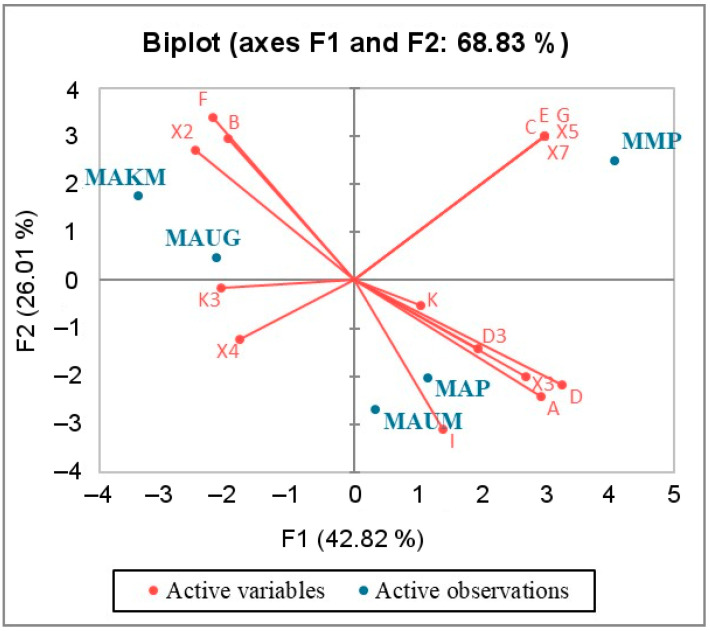
Principal component analysis plot showing degree of similarity among five analyzed groups of modern horses based on the similarity of their mitochondrial haplotype composition. The groups of horses studied here include MAUM (Ulagan “mixed” group), MAUG (Ulagan gene pool group), and MAKM (Kosh-Agach “mixed” group), and the groups of horses described in previous publications [7,10,13,17,34,35] consist of MAP (united Altai horse group) and MMP (united Mongolian horse group).

**Table 1 genes-14-01523-t001:** Haplotypes of modern Altai horses from the studied groups.

Name of Haplogroup	Name of Ulagan Gene Pool Group Haplotypes	Name of Ulagan “Mixed” Group Haplotypes	Name of Kosh-Agach “Mixed” Group Haplotypes	Number of Studied Horses Belonging to Haplogroup
A	A + 652 (5)	A + 652 (3)	–	8
B	B1 (4)	–	B1 (4)	8
D	–	D2 (2), D2 + 596 (1)	–	3
F	F (5), F + 584 (1)	–	F (1)	7
I	–	I (3)	–	3
K	K (4), K2 (8), K2b (5)	K (2), K2 (4), K2b (1)	K2 (1), K2b (1)	26
K3	K3 (21), K3a (4)	K3 (4)	K3 (2)	31
X2	X2 (6), X2b (12)	X2 (2), X2 + 528 (1)	X2 (4), X2 + 521 (3)	28
X3	X3c1 (4)	X3c1 + 632 (2)	–	6
X4	X4a (2)	X4a (4)	X4a (2)	8

(X)—the number of horses. red color—new mitogenome hypervariable region I haplotypes of horses. +XXX—15XXX position in the reference sequence of the horse mitogenome (NC_001640.1).

**Table 2 genes-14-01523-t002:** Assessment results of the genetic diversity of the studied horse groups.

Group Name	Number of Horses	Number of Haplotypes	Haplotype Diversity	Number of Polymorphic Sites	Nucleotide Diversity
MAUM (modern Altai Ulagan “mixed”)	29	12	0.9310 +/− 0.0199	23	0.019609 +/− 0.010986
MAUG (modern Altai Ulagan gene pool)	81	13	0.8846 +/− 0.0197	21	0.018346 +/− 0.010124
MAKM (modern Altai Kosh-Agach “mixed”)	18	8	0.8889 +/− 0.0416	16	0.022064 +/− 0.012482
MAP (modern Altai published)	52	15	0.9201 +/− 0.0166	23	0.019970 +/− 0.010985
MMP (modern Mongolia published)	231	68	0.9633 +/− 0.0042	41	0.019835 +/− 0.010754
AAG (ancient Altai united group)	20	14	0.9579 +/− 0.0281	23	0.018398 +/− 0.010565
AMG (ancient Mongolia united group)	12	8	0.9242 +/− 0.0575	15	0.021664 +/− 0.012679

## Data Availability

The data presented in this study are openly available in [GenBank database], reference numbers are available in Appendix A (ancient samples) and Appendix A (modern samples).

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
