# Peer review of "Genetic History of the Altai Breed Horses: From Ancient Times to Modernity"

_genes, 2023, doi:10.3390/genes14081523_

Round 1

Reviewer 1 Report

The manuscript describes history of the native horse breed of the Siberia based on their genetic markers. In most part it is presented in a clear and understandable way, however, some corrections, including English lanuage are needed.

Specific comments are written below:

- lines 41-42: please, rephrase the sentence, it's not grammatically correct

- line 45: "factory breeds"?

- line 69: "most partial compared", please correct

Results section

- lines 220-225: please, correct; maybe divide in few shorter sentences

- when you give the p-value do not write whether the result is significant or not; p-value itself gives that information, please, check the whole Results section

- lines 333- 335: what groups? it appears that there are different groups from the one described so far (MAUM, MAUG, MAKM, MAP, MMP); this fragment needs to be re-written and clarified. 

Discussion section:

- lines 367-375:please, do not repeat the Introduction, the aim of the study was already given in the previous section (Introduction)

- lines 387-389: this information should be given as a reference after the appropriate fragment in the discussion section

The English language needs to be corrected. 

Author Response

Dear Reviewer,

Thank you very much for your recommendations, comments, suggestions for improving our article. We tried to make all the changes that were suggested. In particular,

Point 1: - lines 41-42: please, rephrase the sentence, it's not grammatically correct

Response 1: The phrase has been reformulated.

Point 2: - line 45: "factory breeds"?

Response 2: We have replaced this term in the article with the term farm breeds, which we consider to be more correct. We used this term to refer to breeds with which in-depth breeding work was carried out for a long time, which consisted in the strict selection of animals, the conditions for their feeding and maintenance.

Point 3: - line 69: "most partial compared", please correct

Response 3: The sentence has been reformulated.

Point 4: - lines 220-225: please, correct; maybe divide in few shorter sentences

Response 4: The sentence was split into two shorter ones.

Point 5: - when you give the p-value do not write whether the result is significant or not; p-value itself gives that information, please, check the whole Results section

Response 5: We have tried to avoid such duplication in the text by removing some information where necessary.

Point 6: - lines 333- 335: what groups? it appears that there are different groups from the one described so far (MAUM, MAUG, MAKM, MAP, MMP); this fragment needs to be re-written and clarified.

Response 6: We have tried to explain the division into groups in more detail.

Point 7: - lines 367-375: please, do not repeat the Introduction, the aim of the study was already given in the previous section (Introduction)

Response 7: We decided to remove this paragraph, since, as rightly noted, this information was already given in the Introduction.

Point 8: - lines 387-389: this information should be given as a reference after the appropriate fragment in the discussion section

Response 8: We have given this information as a reference after the appropriate fragments in the discussion section.

Point 9: The English language needs to be corrected.

Response 9: The English language of the article was edited by a specialist in the field of foreign languages.

Reviewer 2 Report

Dear Authors, it is not clear what does it mean for today: horse breeding today and in the future, for body condition for these described horse breeds?

Author Response

Dear Reviewer,

Thank you very much for your recommendations, comments, suggestions for improving our article. We tried to make all the changes that were suggested. In particular,

Point 1: The presence of a large number of brackets in a paragraph with lines 235-241

Response 1: The sentence was divided into two sentences.

Point 2: the dark green circle is missing in the phylogenetic median-joining network

Response 2: We tried to make the dark green color more visible by changing its hue.

Point 3: replace PCA with principal component analysis

Response 3: We have replaced PCA with principal component analysis in the text of the article.

Point 4: Too many brackets in the text fragment at the end of the section 3.2. Statistical Population Genetics Data Analysis

Response 4: We tried to reduce the number of parentheses by reformulating the sentences in this piece of text.

Point 5: Dear Authors, it is not clear what does it mean for today: horse breeding today and in the future, for body condition for these described horse breeds?

Response 5: We have added a discussion of this question at the end of the Discussion section: “The population genetic analyses carried out (principal component analysis, FST, AMOVA) confirmed the presence of two populations of horses of the Altai breed that live in geographically isolated areas: a mountain population formed in conditions of semi-free living in the subalpine and alpine high-mountain meadows of Ulagan and a steppe pop-ulation formed in the conditions of the semi-desert steppe of Kosh-Agach. Based on the analysis of the mitochondrial DNA control region, we showed that representatives of these populations have not only phenotypic differences [3], but also genetic differences. Differences in these populations may be due to different needs of the inhabitants of these regions: in the mountains, the advantage is given to a drier and stronger horse, which could easily move with riders or cargo over rocky terrain, in the steppe region, horses are more often used to obtain koumiss and meat, therefore their selection was carried out in different directions.

It is possible that the greater genetic similarity of the Ulagan population horses with the Mongolian horses is due to the greater degree of crossing of the Kosh-Agach popula-tion horses with horses of farm breeds or with the peculiarities of their origin, since the ancient horse populations of Altai and Mongolia have significant differences in the gene pool.

Thus, our study opened up prospects for further study of Altai horse populations from different regions, including using other genetic markers.”
